# FlowCam: Training Generalizable 3D Radiance Fields without Camera Poses via Pixel-Aligned Scene Flow

Cameron Smith        Yilun Du        Ayush Tewari        Vincent Sitzmann

{camsmith, yilundu, ayusht, sitzmann}@mit.edu

MIT CSAIL

cameronosmith.github.io/flowcam

## Abstract

Reconstruction of 3D neural fields from posed images has emerged as a promising method for self-supervised representation learning. The key challenge preventing the deployment of these 3D scene learners on large-scale video data is their dependence on precise camera poses from structure-from-motion, which is prohibitively expensive to run at scale. We propose a method that jointly reconstructs camera poses and 3D neural scene representations online and in a single forward pass. We estimate poses by first lifting frame-to-frame optical flow to 3D scene flow via differentiable rendering, preserving locality and shift-equivariance of the image processing backbone. $SE(3)$ camera pose estimation is then performed via a weighted least-squares fit to the scene flow field. This formulation enables us to jointly supervise pose estimation and a generalizable neural scene representation via re-rendering the input video, and thus, train end-to-end and fully self-supervised on real-world video datasets. We demonstrate that our method performs robustly on diverse, real-world video, notably on sequences traditionally challenging to optimization-based pose estimation techniques.

## 1   Introduction

Recent learning-based 3D reconstruction techniques show promise in estimating the underlying 3D appearance and geometry from just a few posed image observations, in a single feed-forward pass [1–19]. These techniques offer an exciting new perspective on computer vision: Instead of making predictions only on pixels, computer vision models might operate directly on the corresponding 3D scenes. This would be a significant step towards a generalist computer vision model, applicable to any task involving interaction with the physical world.

A core challenge to the generality of these methods is that they *cannot* be trained from just video, but instead require knowledge of per-frame camera poses. Existing methods thus rely on curated datasets that obtain camera poses via Structure-from-Motion, but this is prohibitively expensive to compute at scale. Lifting this dataset prerequisite would unlock orders of magnitude more training data, making large-scale 3D representation learning tractable. Meanwhile, odometry and SLAM methods offer online camera pose estimation, but may fail to track sequences with dominant camera rotation or with sparse visual features, and do not reconstruct dense 3D scene representations. While recent efforts leveraging differentiable rendering have demonstrated impressive results at joint reconstruction of camera poses and 3D scenes, they still require minutes or hours per scene. Further, these optimization-based methods cannot leverage learned priors for camera pose estimation, leaving significant progress in computer vision of the last decade untapped. While prior work has demonstrated self-supervised learning of joint depth and camera pose prediction [20, 21], these models are constrained to tight video distributions, such as self-driving video, and do not infer a full 3D representation, only depth.

37th Conference on Neural Information Processing Systems (NeurIPS 2023).

We present a method for jointly training feed-forward generalizable 3D neural scene representation and camera trajectory estimation, self-supervised only by re-rendering losses on video frames, completely without ground-truth camera poses or depth maps. We propose to leverage single-image neural scene representations and differentiable rendering to lift frame-to-frame optical flow to 3D scene flow. We then estimate $SE(3)$ camera poses via a robust, weighted least-squares solver on the scene flow field. Regressed poses are used to re-construct the underlying 3D scene from video frames in a feed-forward pass, where weights are shared with the neural scene representation leveraged in camera pose estimation.

We validate the efficacy of our model for feed-forward novel view synthesis and online camera pose estimation on the real-world RealEstate10K and KITTI datasets, as well as the challenging CO3D dataset. We further demonstrate results on in-the-wild scenes in Ego4D and Walking Tours streamed from YouTube. We demonstrate generalization of camera pose estimation to out-of-distribution scenes and achieve robust performance on trajectories on which a state-of-the-art SLAM approach, ORB-SLAM3 [22], struggles.

To summarize, the contributions of our work include:

- We present a new formulation of camera pose estimation as a weighted least-squares fit of an $SE(3)$ pose to a 3D scene flow field obtained via differentiable rendering.
- We combine our camera pose estimator with a multi-frame 3D reconstruction model, unlocking end-to-end, self-supervised training of camera pose estimation and 3D reconstruction.
- We demonstrate that our method performs robustly across diverse real-world video datasets, including indoor, self-driving, and object-centric scenes.

## 2   Related Work

**Generalizable Neural Scene Representations.**   Recent progress in neural fields [23–25] and differentiable rendering [10, 26–31] have enabled novel approaches to 3D reconstruction from few or single images [1–19], but require camera poses both at training and test time. An exception is recently proposed RUST [32], which can be trained for novel view synthesis without access to camera poses, but does not reconstruct 3D scenes explicitly and does not yield explicit control over camera poses. We propose a method that is similarly trained self-supervised on real video, but yields explicit camera poses and 3D scenes in the form of radiance fields. We outperform RUST on novel view synthesis and demonstrate strong out-of-distribution generalization by virtue of 3D structure.

**SLAM and Structure-from-Motion (SfM).**   SfM methods [33–35], and in particular, COLMAP [34], are considered the de-facto standard approach to obtaining accurate geometry and camera poses from video. Recent progress on differentiable rendering has enabled joint estimation of radiance fields and camera poses via gradient descent [36–40], enabling subsequent high-quality novel view synthesis. Both approaches require offline per-scene optimization. In contrast, SLAM methods usually run online [22, 41, 42], but are notoriously unreliable on rotation-heavy trajectories or scenes with sparse visual features. Prior work proposes differentiable SLAM to learn priors over camera poses and geometry [43, 44], but requires ground-truth camera poses for training. Recent work has also explored how differentiable rendering may be directly combined with SLAM [45–48], usually using a conventional SLAM algorithm as a backbone and focusing on the single-scene overfitting case. We propose a fully self-supervised method to train generalizable neural scene representations without camera poses, outperforming prior work on generalizable novel view synthesis without camera poses. We do *not* claim state-of-the-art camera pose estimation, but provide an analysis of camera pose quality nevertheless, demonstrating robust performance on sequences that are challenging to state-of-the-art SLAM algorithms, ORB-SLAM3 [22] and Droid-SLAM [43].

**Neural Depth and Camera Pose Estimation.**   Prior work has demonstrated joint self-supervised learning of camera pose and monocular depth [20, 21, 49–51] or multi-plane images [52]. These approaches leverage a neural network to *directly* regress camera poses with the primary goal of training high-quality monocular depth predictors. They are empirically limited to sequences with simple camera trajectories, such as self-driving datasets, and do not enable dense, large-baseline novel view synthesis. We ablate our flow-based camera pose estimation with a similar neural network-based approach. Most closely related to our work are approaches that infer per-timestep 3D voxel grids and train a CNN to regress frame-to-frame poses [53, 54]. We benchmark with the most recent approach in this line of work, Video Autoencoder [53]. Lastly, we strongly encourage the reader to peruse impressive concurrent work DBARF [55], which also regresses camera poses alongside a generalizable NeRF. Key differences are that we leverage a pose solver based on 3D-lifted optical

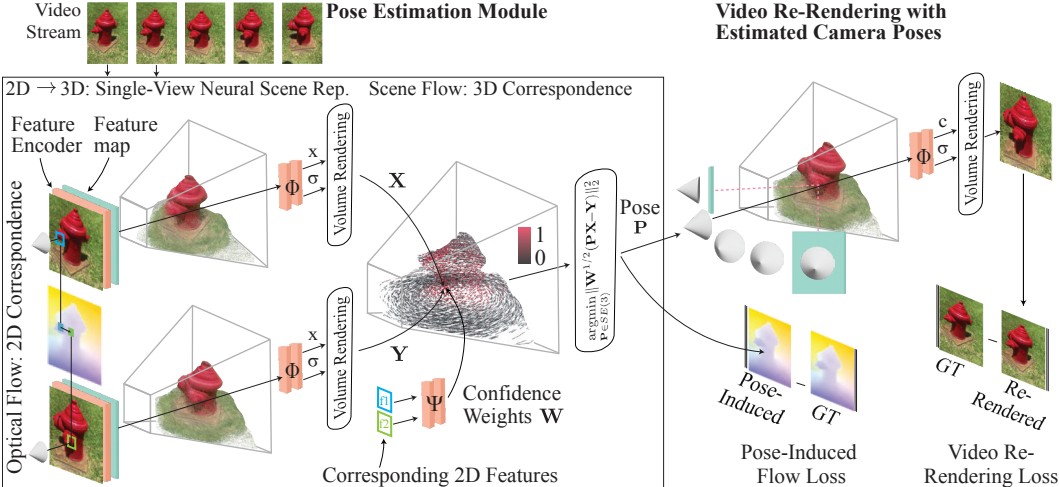

Figure 1: **Method Overview.** Given a set of video frames, our method first computes frame-to-frame camera poses (left) and then re-renders the input video (right). To estimate pose between two frames, we compute off-the-shelf optical flow to establish 2D correspondences. Using single-view pixelNeRF [1], we obtain a surface point cloud as the expected 3D ray termination point for each pixel, $\mathbf{X}$, $\mathbf{Y}$ respectively. Because $\mathbf{X}$ and $\mathbf{Y}$ are pixel-aligned, optical flow allows us to compute 3D scene flow as the difference of corresponding 3D points. We then find the camera pose $\mathbf{P} \in \mathrm{SE}(3)$ that best explains the 3D flow field by solving a weighted least-squares problem with flow confidence weights $\mathbf{W}$. Using all frame-to-frame poses, we re-render all frames. We enforce an RGB loss and a flow loss between projected pose-induced 3D scene flow and 2D optical flow. Our method is trained end-to-end, assuming only an off-the-shelf optical flow estimator.

flow for real-time odometry versus predicting iterative updates to pose and depth via a neural network. Further, we extensively demonstrate our method's performance on rotation-dominant video sequences, in contrast to a focus on forward-facing scenes. Lastly, we solely rely on the generalizable scene representation in contrast to leveraging a monocular depth model for pose estimation.

# 3 Learning 3D Scene Representations from Unposed Videos

Our model learns to map a monocular video with frames $\{\mathbf{I}_t\}_{t=1}^N$ as well as off-the-shelf optical flow $\{\mathbf{V}_t\}_{t=1}^{N-1}$ to per-frame camera poses $\{\mathbf{P}_t\}_{t=1}^N$ and a 3D scene representation $\Phi$ in a single feed-forward pass. We leverage known intrinsic parameters when available, but may predict them if not. We will first introduce the generalizable 3D scene representation $\Phi$. We then discuss how we leverage $\Phi$ for feed-forward camera pose estimation, where we lift optical flow into 3D scene flow and solve for pose via a weighted least-squares $\mathrm{SE}(3)$ solver. Finally, we discuss how we obtain supervision for both the 3D scene representation and pose estimation by re-rendering RGB and optical flow for all frames. An overview of our method is presented in Fig. 1.

**Notation.** It will be convenient to treat images sometimes as discrete tensors, such as $\mathbf{I}_t \in \mathbb{R}^{H \times W \times 3}$, and sometimes as functions $I : \mathbb{R}^2 \to \mathbb{R}^3$ over 2D pixel coordinates $\mathbf{p} \in \mathbb{R}^2$. We will denote functions in italic $I$, while we denote the corresponding tensors sampled on the pixel grid in bold as $\mathbf{I}$.

## 3.1 Defining Our Image-Conditioned 3D Scene Representation

First, we introduce the generalizable 3D scene representation we aim to train. Our discussion assumes known camera poses; in the subsequent section we will describe how we can use our scene representation to estimate them instead. We parameterize our 3D scene as a Neural Radiance Field (NeRF) [28], such that $\Phi$ is a function that maps a 3D coordinate $\mathbf{x}$ to a color $\mathbf{c}$ and density $\sigma$ as $\Phi(\mathbf{x}) = (\sigma, \mathbf{c})$. To render the color for a ray $\mathbf{r}$, points $(\mathbf{x}_1, \mathbf{x}_2, ..., \mathbf{x}_n)$ are sampled along $\mathbf{r}$ between predefined near and far planes $[t_1, t_f]$, fed into $\Phi$ to produce corresponding color and density tuples $[(\sigma_1, \mathbf{c}_1), (\sigma_2, \mathbf{c}_2), ..., (\sigma_n, \mathbf{c}_n)]$, and alpha-composited to produce a final color value $C(\mathbf{r})$:

$$C(\mathbf{r}) = \sum_{i=1}^N T_i(1 - \exp(-\sigma_i \delta_i))\mathbf{c}_i, \text{ where } T_i = \exp(-\sum_{j=1}^{i-1} \sigma_j \delta_j), \tag{1}$$

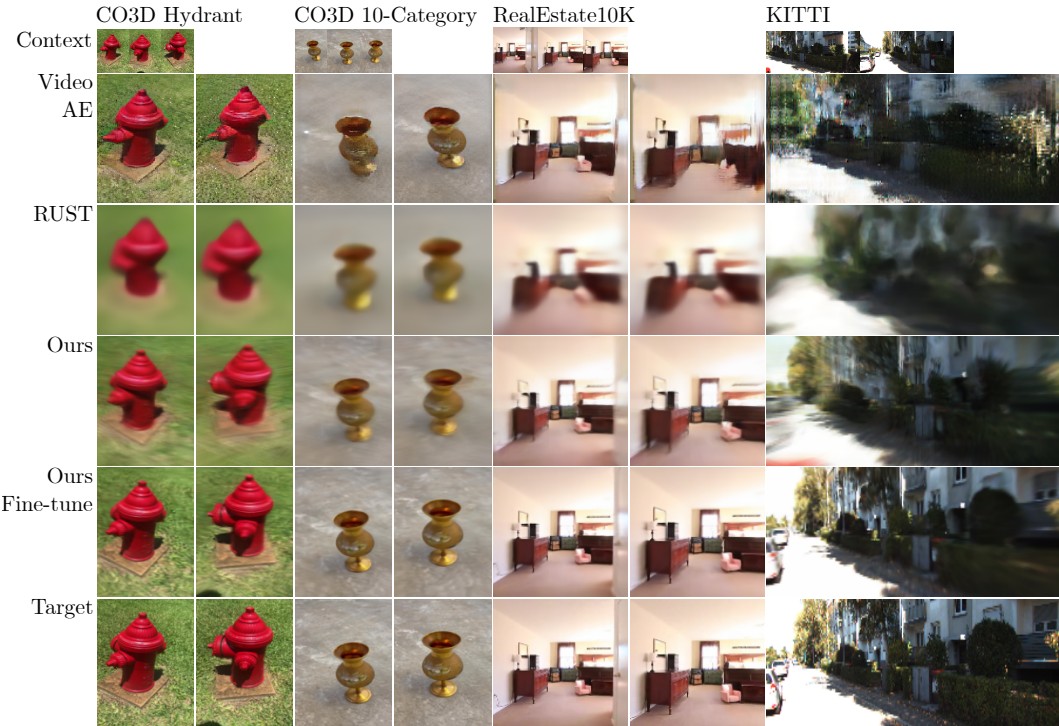

Figure 2: **Video Reconstruction Results.** Our model reconstructs video frames from sparse context frames with higher fidelity than all baselines. While VidAE's renderings often appear with convincing texture, they are often not aligned with the ground truth. RUST's renderings are well aligned but are blurry due to their coarse set latent representation.

| Model | CO3D-Hydrants | | CO3D-10 | | RealEstate | | KITTI | |
|---|---|---|---|---|---|---|---|---|
| | LPIPS ↓ | PSNR ↑ | LPIPS ↓ | PSNR ↑ | LPIPS ↓ | PSNR ↑ | LPIPS ↓ | PSNR ↑ |
| Vid-AE [53] | 0.3427 | 18.56 | 0.3889 | 18.50 | 0.4173 | 18.03 | 0.3272 | 17.65 |
| RUST [32] | 0.6126 | 17.50 | 0.6145 | 17.71 | 0.5692 | 18.07 | 0.5584 | 15.44 |
| Ours | **0.2250** | **23.94** | **0.2687** | **24.17** | **0.2224** | **24.25** | **0.1928** | **23.39** |

Table 1: **Quantitative Comparison on View Synthesis.** On the task of view synthesis, our method outperforms other unposed methods by wide margins.

where $\delta_i = t_{i+1} - t_i$ is the distance between adjacent samples. By compositing sample locations instead of colors, we can compute an expected ray-surface intersection point $S(\mathbf{r}) \in \mathbb{R}^3$:

$$S(\mathbf{r}) = \sum_{i=1}^{N} T_i (1 - \exp(-\sigma_i \delta_i)) \mathbf{x}_i. \tag{2}$$

We require a *generalizable* NeRF that is not optimized for each scene separately, but instead predicted in a feed-forward pass by an encoder that takes a set of $M$ context images and corresponding camera poses $\{(\mathbf{I}_i, \mathbf{P}_i)\}_i^M$ as input. We denote such a generalizable radiance field reconstructed from images $\mathbf{I}_i$ as $\Phi(\mathbf{x} \mid \{(\mathbf{I}_i, \mathbf{P}_i)\}_i^M)$. Many such models have been proposed [1–10]. We base our model on pixelNeRF [1], which we briefly discuss in the following - please find further details in the supplement. pixelNeRF first extracts per-image features $\mathbf{F}_i$ from each input image $\mathbf{I}_i$. A given 3D coordinate $\mathbf{x}$ is first projected onto the image plane of each context image $\mathbf{I}_i$ via the known camera pose and intrinsic parameters to yield pixel coordinates $\mathbf{p}_i$. We then retrieve the features $F_i(\mathbf{p}_i)$ at that pixel coordinate. Color and density $(\sigma, \mathbf{c})$ at $\mathbf{x}$ are then predicted by a neural network that takes as input the features $\{F_i(\mathbf{p}_i)\}_i^M$ and the coordinates of $\mathbf{x}$ in the coordinate frame of each camera, $\{\mathbf{P}_i^{-1}\mathbf{x}\}_i^M$. Importantly, we can condition pixelNeRF on varying numbers of context images, i.e., we may run pixelNeRF with only a *single* context image as $\Phi(\mathbf{x} \mid (\mathbf{I}, \mathbf{P}))$, or with a set of $M > 1$ context images $\Phi(\mathbf{x} \mid \{(\mathbf{I}_i, \mathbf{P}_i)\}_i^M)$.

### 3.2 Lifting Optical Flow to Scene Flow with Neural Scene Representations

Equipped with our generalizable 3D representation $\Phi$, we now describe how we utilize it to lift optical flow into confidence-weighted 3D scene flow. Later, our pose solver will fit a camera pose to the

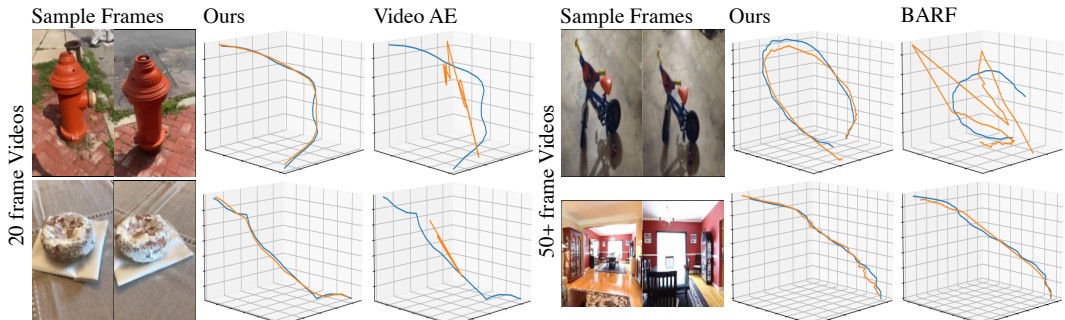

Figure 3: **Qualitative Pose Estimation Comparison**. On short sequences, we compare our pose estimation to Video Autoencoder [53], and on long sequences, we compare our method's sliding-window estimations against the per-scene optimization BARF [38]. The trajectory for the bicycle sequence was obtained using a model trained on hydrant sequences: despite never having seen a bicycle before, our model predicts accurate poses.

(a)

| | Hydrant | 10-Cat. | RE10K | KITTI |
|---|---|---|---|---|
| VidAE [53] | 0.100 | 0.101 | 0.090 | 0.084 |
| Ours | **0.041** | **0.042** | **0.020** | **0.010** |

(b)

| | Top | Bot. | % Tracked |
|---|---|---|---|
| ORB3 [22] | 0.021 | 0.034 | 49 |
| DROID [43] | 0.028 | 0.049 | **100** |
| Ours | **0.015** | **0.025** | **100** |

Table 2: **Quantitative Pose Estimation Comparison.** In **(a)** we compare against VideoAutoencoder [53] on short-sequence odometry estimation (20 frames), reporting the ATE. In **(b)** we compare against ORB-SLAM3 [22] and DROID-SLAM [43] on long sequences (∼200 frames) from the CO3D 10-Category dataset. We separately report scores on the top and bottom 50% of sequences ("Top" and "Bot.") in terms of quality of ground-truth poses as indicated by the dataset. We report ATE and percent of sequences tracked ("Tracked"). ORB-SLAM3 fails to track over half of these challenging sequences.

estimated scene flow. Given two sequential frames $\mathbf{I}_{t-1}, \mathbf{I}_t$ we first use an off-the-shelf method [56] to estimate backwards optical flow $V_t : \mathbb{R}^2 \rightarrow \mathbb{R}^2$. The optical flow $V_t$ maps a 2D pixel coordinate $\mathbf{p}$ to a 2D flow vector, such that we can determine the corresponding pixel coordinate in $\mathbf{I}_{t-1}$ as $\mathbf{p}' = \mathbf{p} + V_t(\mathbf{p})$.

We will now lift pixel coordinates $\mathbf{p}_t$ and $\mathbf{p}_{t-1}$ to an estimate of the 3D points that they observe in the coordinate frame of their respective cameras. To achieve this, we cast rays from the camera centers through the corresponding pixel coordinates $\mathbf{p}_t$ and $\mathbf{p}_{t-1}$ using the intrinsics matrix $\mathbf{K}$. Specifically, we compute $\mathbf{r}_t = \mathbf{K}^{-1}\tilde{\mathbf{p}}_t$ and $\mathbf{r}_{t-1} = \mathbf{K}^{-1}\tilde{\mathbf{p}}_{t-1}$, where $\tilde{\mathbf{p}}$ represents the homogeneous coordinate $\binom{\mathbf{P}}{1}$. Next, we sample points along the rays $\mathbf{r}_t$ and $\mathbf{r}_{t-1}$ and query our pixelNeRF model in the single-view setting. This involves invoking $\Phi(\cdot|(\mathbf{I}_t, \mathbb{I}_{4\times4}))$ and $\Phi(\cdot|(\mathbf{I}_{t-1}, \mathbb{I}_{4\times4}))$, i.e., pixelNeRF is run with only the respective frame as the context view and the identity matrix $\mathbb{I}$ as the camera pose. Applying the ray-intersection integral defined in Eq. 2 to the pixelNeRF estimates, we obtain a pair of 3D points $(\mathbf{x}_t, \mathbf{x}_{t-1})$ corresponding to the estimated surfaces observed by pixels $\mathbf{p}_t$ and $\mathbf{p}_{t-1}$, respectively. We repeat this estimation for all optical flow correspondences, resulting in two sets of surface point clouds, $\mathbf{X}, \mathbf{X}' \in \mathbb{R}^{H\times W\times 3}$. Equivalently, we may view this as defining the 3D scene flow as $\mathbf{X}' - \mathbf{X}$.

**Flow confidence weights.** We further utilize a confidence weight for each flow correspondence. To accomplish this, we employ a neural network $\Psi$, which takes image features $F_t(\mathbf{p}), F_{t-1}(\mathbf{p}')$ as input for every pixel correspondence pair $(\mathbf{p}, \mathbf{p}')$. The network maps these features to a weight $\mathbf{w}$, denoted as $\Psi(F_t(\mathbf{p}), F_{t-1}(\mathbf{p}')) = \mathbf{w} \in [0, 1]$. $\Psi$ can importantly overcome several failure modes which lead to faulty pose estimation, including incorrect optical flow, such as in areas of occlusions, dynamic objects, such as pedestrians, or challenging geometry estimates, such as sky regions. We show in Fig. 9 that $\Psi$ indeed learns such content-based rules.

**Predicting Intrinsic Camera Parameters K.** Camera intrinsics are often approximately known, either published by the manufacturer, saved in video metadata, or calibrated once. Nevertheless, for purposes of large-scale training, we leverage a simple scheme to predict the camera field-of-view for a video sequence. We feed the feature map of the first frame $\mathbf{F}_0$ into a convolutional encoder that directly regresses the field of view. We assume that the optical center is at the sensor center. We find that this approach enables us to train on real-world video from YouTube, though more sophisticated schemes [57] will no doubt improve performance further.

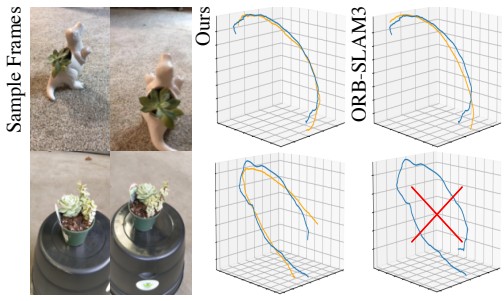
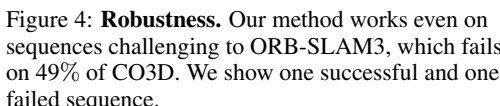
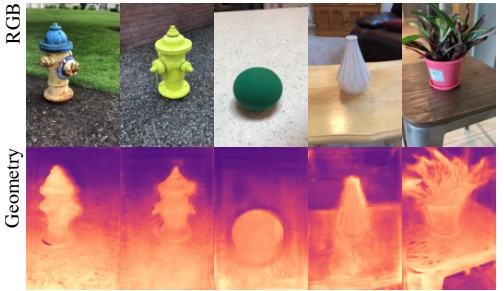

Figure 4: **Robustness.** Our method works even on sequences challenging to ORB-SLAM3, which fails on 49% of CO3D. We show one successful and one failed sequence.

Figure 5: **Learned geometry.** Our model's expected ray termination illustrates the unsupervised geometry learned by our model on the challenging CO3D dataset.

## 3.3 Camera Pose Estimation as Explaining the Scene Flow Field

We will now estimate the camera pose between frame $\mathbf{I}_t$ and $\mathbf{I}_{t-1}$. In the previous section, we lifted the input optical flow into scene flow, producing 3D correspondences $\mathbf{X}, \mathbf{X}'$, or, equivalently, 3D scene flow. We cast camera pose estimation as the problem of finding the rigid-body motion that best explains the observed scene flow field, or the transformation mapping points in $\mathbf{X}$ to $\mathbf{X}'$, while considering confidence weights $\mathbf{W}$. Note that below, we will refer to the matrices $\mathbf{X}, \mathbf{X}'$, and $\mathbf{W}$ as column vectors, with their spatial dimensions flattened.

We use a weighted Procrustes formulation to solve for the rigid transformation that best aligns the set of points $\mathbf{X}$ and $\mathbf{X}'$. The standard orthogonal Procrustes algorithm solves for the $\mathrm{SE}(3)$ pose such that it minimizes the least squares error:

$$\underset{\mathbf{P} \in \mathrm{SE}(3)}{\arg\min} \|\tilde{\mathbf{X}} - \mathbf{P}\tilde{\mathbf{X}}'\|_2^2, \tag{3}$$

with $\mathbf{P} = \left(\begin{smallmatrix} \mathbf{R} & \mathbf{t} \\ 0 & 1 \end{smallmatrix}\right)$ as a rigid-body pose with rotation $\mathbf{R}$ and translation $\mathbf{t}$ and homogeneous $\tilde{\mathbf{X}} = \left(\begin{smallmatrix} \mathbf{X} \\ 1 \end{smallmatrix}\right)$. In other words, the minimizer of this loss is the rigid-body transformation that best maps $\mathbf{X}$ onto $\mathbf{X}'$, and, in a static scene, is therefore equivalent to the sought-after camera pose.

As noted by Choy et al. [58], this formulation equally weights all correspondences. As noted in the previous section, however, this would make our pose estimation algorithm susceptible to both incorrect correspondences as well as correspondences that should be down-weighted by nature of belonging to parts of the scene that are specular, dynamic, or have low confidence in their geometry estimate. Following [58], we thus minimize a *weighted* least-squares problem:

$$\underset{\mathbf{P} \in \mathrm{SE}(3)}{\arg\min} \|\mathbf{W}^{1/2}(\tilde{\mathbf{X}} - \mathbf{P}\tilde{\mathbf{X}}')\|_2^2 \tag{4}$$

with the diagonal weight matrix $\mathbf{W} = \mathrm{diag}(\mathbf{w})$. Conveniently, this least-squares problem admits a closed-form solution, efficiently calculated via Singular Value Decomposition, as derived in [58]:

$$\mathbf{R} = \mathbf{U}\mathbf{S}\mathbf{V}^T \text{ and } \mathbf{t} = (\mathbf{X} - \mathbf{R}\mathbf{X}')\mathbf{W}\mathbf{1}, \text{ where } \mathbf{U}\boldsymbol{\Sigma}\mathbf{V}^T = \mathrm{SVD}(\Sigma_{\mathbf{X}'\mathbf{X}}), \tag{5}$$

$$\Sigma_{\mathbf{X}'\mathbf{X}} = \mathbf{X}\mathbf{K}\mathbf{W}\mathbf{K}\mathbf{X}'^T, \mathbf{K} = \mathbb{I} - \sqrt{\mathbf{w}}\sqrt{\mathbf{w}}^T, \text{ and } \mathbf{S} = \mathrm{diag}(1, ..., \det(\mathbf{U})\det(\mathbf{V})). \tag{6}$$

**Composing frame-to-frame poses.** Solving this weighted least-squares problem for each subsequent frame-to-frame pair yields camera transformations $(\mathbf{P}'_2, \mathbf{P}'_3, \ldots, \mathbf{P}'_n)$, aligning each $\mathbf{I}_t$ to its predecessor $\mathbf{I}_{t-1}$. We compose frame-to-frame transformations to yield camera poses $(\mathbf{P}_1, \mathbf{P}_2, \ldots, \mathbf{P}_n)$ relative to the first frame, such that $\mathbf{P}_1 = \mathbb{I}_{3\times3}$, concluding our camera pose estimation module.

## 3.4 Supervision via Differentiable Video and Flow Re-Rendering

We have discussed our generalizable neural scene representation $\Phi$ and our camera pose estimation module. We will now discuss how we derive supervision to train both modules end-to-end. We have two primary loss signals: First, a photometric loss $\mathcal{L}_{\mathrm{RGB}}$ scores the visual fidelity of re-rendered video frames. Second, a pose-induced flow loss $\mathcal{L}_{\mathrm{pose}}$ scores how similar the flow induced by the

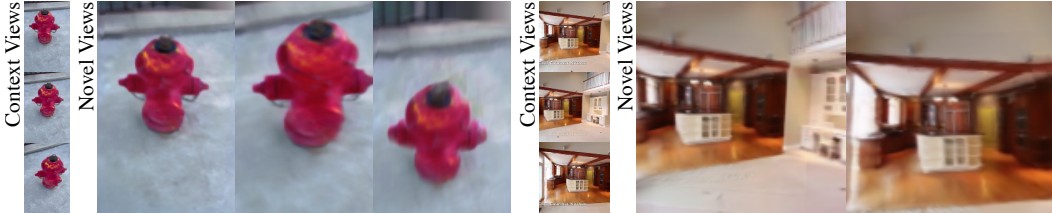

Figure 6: **Wide-Baseline View Synthesis.** Given an input video without poses, our model first infers camera poses and can then render wide-baseline novel views of the underlying 3D scene, where we use the first, middle, and final frame of the video as context views.

predicted camera transformations and surface estimations are to optical flow estimated by RAFT [59]. Our model is trained on short ($\sim$15 frames) video sequences.

**Photometric Loss.** Our photometric loss $\mathcal{L}_{\text{RGB}}$ comprises two terms: A multi-context loss and a single-context loss. The multi-context loss ensures that the full 3D scene is reconstructed accurately. Here, we re-render each frame of the input video sequence $\mathbf{I}_t$, using its estimated camera pose $\mathbf{P}_t$ and multiple context images $\{(\mathbf{I}_j, \mathbf{P}_j)\}_j^J$. The single-context loss ensures that the single-context pixelNeRF used to estimate surface point clouds $\mathbf{X}_t$ in the pose estimation module is accurate.

$$\mathcal{L}_{\text{RGB}} = \frac{1}{N} \sum_{i=t}^{N} \underbrace{\left\| \mathbf{I}_t - C(\mathbf{P}_t \mid \{(\mathbf{I}_j, \mathbf{P}_j)\}_j^J) \right\|_2^2}_{\text{Multi-Context Loss}} + \underbrace{\left\| \mathbf{I}_t - C(\mathbb{I}_{4\times4} \mid (\mathbf{I}_t, \mathbb{I}_{4\times4})) \right\|_2^2}_{\text{Single-Context Loss}}, \quad (7)$$

where, in a slight abuse of notation, we have overloaded the rendering function $C(\mathbf{P}|\{(\mathbf{I}_j, \mathbf{P}_j)\}_j^J)$ defined in Eq. 1 as rendering out the full image obtained by rendering a pixelNeRF with context images $\{(\mathbf{I}_j, \mathbf{P}_j)\}_j^J$ from camera pose $\mathbf{P}$. $N$ refers to the number of frames in the video sequence and $J$ refers to the subset of context frames use for re-rendering the entire sequence. We first attempted picking the first frame only, however, found that this does *not* converge due to the uncertainty of the 3D scene given only the first frame: single-view pixelNeRF will generate blurry estimates for parts of the scene that have high uncertainty, such as occluded regions or previously unobserved background.

**Pose-Induced Flow Loss.** An additional, powerful source of supervision for both the estimated geometry and camera poses can be obtained by comparing the optical flow induced by the predicted surface point clouds and pose with the off-the-shelf optical flow. We define this pose-induced flow loss as

$$\mathcal{L}_{\text{pose}} = \frac{1}{N-1} \sum_{t=1}^{N-1} \left\| \mathbf{V}_t - (\pi(\mathbf{P}_t^{-1} \cdot \mathbf{P}_{t+1} \cdot \mathbf{X}_{t+1}) - \mathbf{uv}) \right\|_2^2, \quad (8)$$

with projection operator $\pi(\cdot)$ and grid of pixel coordinates $\text{uv} \in \mathbb{R}^2$. Intuitively, this transforms the surface point cloud of frame $t + 1$ into the coordinate frame of frame $t$ and projects it onto that image plane. For every pixel coordinate $\mathbf{p}$ at timestep $t + 1$, this yields a corresponding pixel coordinate $\mathbf{p}'$ at timestep $t$, which we compare against the input optical flow.

### 3.5 Test-time Inference

After training our model on a large dataset of short video sequences, we may infer both camera poses and a radiance field of such a short sequence in a single forward pass, without test-time optimization.

**Sliding Window Inference for Odometry on Longer Trajectories.** Our method estimates poses for short ($\sim$15 frames) subsequences in a single feed-forward pass. To handle longer trajectories that exceed the capacity of a single batch, we divide a given video into non-overlapping subsequences. We estimate poses for each subsequence individually and compose them to create an aggregated trajectory estimate. This approach allows us to estimate trajectories for longer video sequences.

**Test-Time Adaptation.** Frame-to-frame camera pose estimation methods, both conventional and the proposed method, accumulate pose estimation error over the course of a sequence. SLAM and SfM methods usually have a mechanism to correct for drift by globally optimizing over all poses and closing loops in the pose graph [60]. We do not have such a mechanism, but propose fine-tuning our model on specific scenes for more accurate feed-forward pose and 3D estimation. For a given video sequence, we may fine-tune our pre-trained model using our standard photometric and flow losses on sub-sequences of the video. Note that this is *not* equivalent to per-scene optimization or *direct*

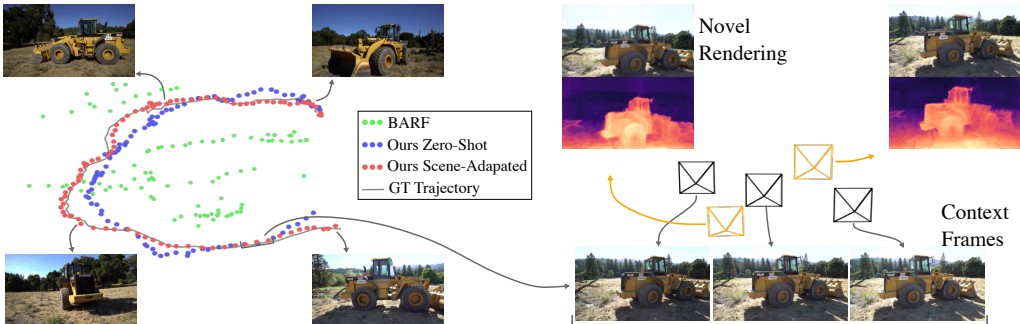

Figure 7: **Fine-tuned Pose Estimation and View Synthesis on Large-Scale, Out-of-Distribution Scene.** We evaluate our RealEstate10K-trained model on a significantly out-of-distribution scene from the Tanks and Temples dataset [61], first without any scene-adaptation and then with. Even with this significant distribution gap, our method's estimated trajectory captures the looping structure of the ground truth, albeit with accumulated drift. After a scene-adaptation fine-tuning stage (around 7 hours), our model estimates poses which align closely with the ground truth. We also plot the trajectory estimated by BARF [38], which fails to capture the correct pose distribution.

optimization of camera poses and a radiance field, as performed e.g. in BARF [38]: neither camera poses nor the radiance field are free variables. Instead, we fine-tune the weights of our convolutional inference backbone and MLP renderer for more accurate feed-forward prediction.

## 4   Experiments

We benchmark our method on generalizable novel view synthesis on the RealEstate10k [62], CO3D [63], and KITTI [64] datasets. We provide further analysis on the Tanks & Temples dataset and in-the-wild scenes from Ego4D [65] and YouTube. Though we do not claim to perform state-of-the-art camera pose estimation, we nevertheless provide an analysis of the accuracy of our estimated camera poses. Please find more results, as well as precise hyperparameters, implementation, and dataset details, in the supplemental document and video. We utilize camera intrinsic parameters where available, predicting them only for the in-the-wild Ego4D and WalkingTours experiments.

**Pose Estimation.**   We first evaluate our method on pose estimation against the closest self-supervised neural network baseline, Video Autoencoder (VidAE) [53]. We then analyze the robustness of our pose estimation with ORB-SLAM3 [66] and DROID-SLAM [43] as references. Finally, we benchmark with BARF [38], a single-scene unposed NeRF baseline. Tab. 2a compares accuracy of estimated poses of our method and VidAE on all four datasets. The performance gap is most pronounced on the challenging CO3D dataset, but even on simpler, forward-moving datasets, RealEstate10k and KITTI, our method significantly outperforms VidAE. Next, we analyse the robustness of our pose estimation module on CO3D, using SfM methods ORB-SLAM3 and DROID-SLAM as references. See Tab. 2b and Fig. 4 for quantitative and qualitative results. To account for inaccuracies in the provided CO3D poses we utilize as ground-truth, we additionally report separate results for the top and bottom 50% of sequences, ranked based on the pose confidence scores provided by the authors. Although we do not employ any secondary pose method as a proxy ground truth for the bottom half of sequences, this division serves as an approximate indication of the level of difficulty each sequence poses from a SfM perspective. On both subsets, our method outperforms both DROID-SLAM and ORB-SLAM3. Also note that ORB-SLAM3 fails to track poses for over half (50.7%) of the sequences. On the sequences where ORB-SLAM3 succeeds, our method predicts poses significantly more accurately. Even on the sequences where ORB-SLAM3 fails, our performance does not degrade (.025 ATE). Lastly, we compare against the single-scene unposed NeRF baseline, BARF. Since BARF requires ∼one day to reconstruct a single sequence, we evaluate on two representative sequences: a forward-walking sequence on RealEstate10K, and an outside-in trajectory on CO3D. We plot recovered trajectories in Fig. 3. While BARF fails to recover the correct trajectory shape on the CO3D scene, our method produces a trajectory that more accurately reflects the ground-truth looping structure.

**Novel View Synthesis.**   We compare against VidAE [53] and RUST [32] on the task of novel view synthesis. Tab. 1 and Fig. 6 report quantitative and qualitative results respectively. Our method outperforms both baselines significantly. Since VidAE fails to capture the pose distribution on the CO3D datasets, its novel view renderings generally do not align with the ground truth. On RealEstate10K and KITTI, their method successfully captures the pose distribution, but still struggles

| Sample Frames | Novel View/ Novel Depth | Sample Frames | Novel View/ Novel Depth | Input RGB | Flow-Weight Masked RGB | Input RGB | Flow-Weight Masked RGB |

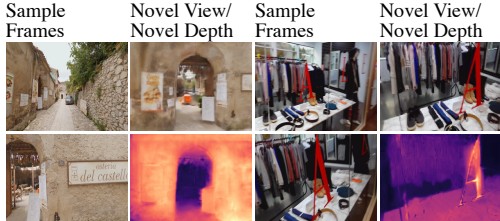 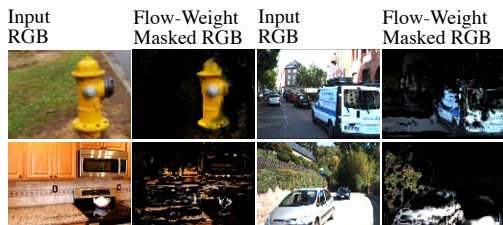

Figure 8: **View Synthesis on Ego4D and Walking Tours**: We train on a collection of YouTube walking tour videos and Ego4D sequences with unknown camera parameters, and render novel views after a short fine-tuning stage.

Figure 9: **Flow Weights**: Our flow-confidence weights allow our model to down-weight unreliable flow correspondences due to occlusions, specular highlights, or dynamic objects, and up-weight well-textured regions.

to render high-fidelity images. We similarly outperform RUST, which struggles to capture high-frequency content due to its coarse scene representation. Also note that RUST estimates latent camera poses rather than explicit ones. We further qualitatively evaluate our method on wide-baseline novel view synthesis, where no ground-truth is available - please see Fig. 6 for these results.

**Test-Time Adaptation**  We evaluate the ability of our model to estimate camera poses on out-of-distribution scenes, augmented by test-time optimization as discussed in section 3.5. Fig.7 displays the camera poses estimated by our method on the Bulldozer sequence from the Tanks and Temples dataset[61], using our model trained on the RealEstate10k dataset. Even without any test-time adaptation, our method generalizes impressively, generating a trajectory that follows the outside-in trajectory as determined by COLMAP. We refer to this generalization mode as "zero-shot." Nevertheless, camera poses are not perfect and exhibit some drift. We thus perform test-time optimization, fine-tuning the weights of our feed-forward method on subsequences of the video. Though geometry is plausible after roughly 10 minutes of fine-tuning, we continued optimization for 7 hours for finer details. Our scene-adapted estimates are close to ground truth with little drift, and we find that novel view synthesis results with significant baseline are visually compelling with sound depth estimates. We further show the result of BARF on this sequence, which fails to recover a plausible trajectory.

**Results on In-the-Wild Video.**  We present preliminary results on the Ego4D dataset [65], a recent ego-centric dataset captured with headset cameras, and a new Walking Tours dataset, in which we stream walking tour videos from YouTube. Both datasets comprise some scene motion, such as pedestrians, cars, or people. Images from Ego4D are further subject to radial camera distortion, which we do not model. Nevertheless, after fine-tuning on Walking Tours and Ego4D for 20 minutes and 1.5 hours, we can generate plausible novel views and depth maps, illustrated in Fig. 8. We find that our model is generally robust to dynamic scene content, such as pedestrians or humans, which obtain low flow-confidence flow scores. We further run COLMAP on a subset of the Walking Tours and Ego4D videos, yielding pseudo ground-truth poses for 3 videos from each dataset. The Walking Tour subset is selected randomly, and the Ego4D subset is chosen as the first three relatively-static videos we found (to ensure COLMAP convergence). We divide each video into subsequences of 20 frames at 3fps, corresponding to roughly 7 meters of forward translation per clip. Here, we achieve a competitive ATE of 0.013 and 0.026 on Walking Tours and Ego4D, respectively. We also compute PSNR values on subsequences of 10 frames using two context views, and obtain 18.87dB and 21.59db on Walking Tours and Ego4D, respectively, indicating generally plausible novel view synthesis.

|  | ↑ PSNR | ↓ LPIPS |
|---|---|---|
| MLP-Pose | 18.50 | 0.54 |
| No Flow Weights | 17.87 | 0.51 |
| Full | **21.02** | **0.38** |

Table 3: **Ablation study.**

**Ablations.**  In Tab. 3, we ablate key contributions related to our pose formulation, evaluated on the CO3D Hydrant dataset. We first ablate our proposed flow-based pose formulation in favor of concatenating adjacent frames and predicting a pose directly via a CNN, as is common in the Monodepth [67] line of work. We further ablate our weighted flow formulation in favor of a non-weighted Procrustes estimation. Both methods perform significantly worse than our full flow-based pose formulation, and qualitatively often lead to degenerate geometry estimates. Please see the supplemental material for additional ablations and ablation details.

# 5 Discussion

**Limitations.** While we believe our method makes significant strides, it still has several limitations. As an odometry method, it accumulates drift and has no loop closure mechanism. Our model further does not currently incorporate scene dynamics, but recent advancements in dynamic NeRF papers [68–70] present promising perspectives for future research.

**Conclusion.** We have introduced FlowCam, a model capable of regressing camera poses and reconstructing a 3D scene in a single forward pass from a short video. Our key contribution is to factorize camera pose estimation as first lifting optical flow to pixel-aligned scene flow via differentiable rendering, and then solving for camera pose via a robust least-squares solver. We demonstrate the efficacy of our approach on a variety of challenging real-world datasets, as well as in-the-wild videos. We believe that they represent a significant step towards enabling scene representation learning on uncurated, real-world video.

**Acknowledgements.** This work was supported by the National Science Foundation under Grant No. 2211259, by the Singapore DSTA under DST00OECI20300823 (New Representations for Vision), and by the Amazon Science Hub. The Toyota Research Institute also partially supported this work. This article solely reflects the opinions and conclusions of its authors and no other entity.

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
