# OpenReview forum: "FlowCam: Training Generalizable 3D Radiance Fields without Camera Poses via Pixel-Aligned Scene Flow"
_NeurIPS.cc/2023/Conference — NeurIPS 2023 poster_

### Official Review · Reviewer_52at · 2023-06-29

**Soundness:** 2 fair
**Presentation:** 3 good
**Contribution:** 2 fair
**Rating:** 6
**Confidence:** 5

**Summary:**

This paper presents a generalizable framework for estimating both the NeRF model of the scene and the camera pose for video sequence simultiunusly.

The proposed method first uses the pretrained model to predict the optical flow for each video frame. Then, the optical flow is lifted up to the monodepth to scene flow. The camera poses are solved by SVD with scene flow. After computing the camera pose, the nerf model is further fine-tuned for a better result.

**Strengths:**

- a light step of unposed nerf in RGB video sequence setting.
- the quality of the rendered RGB image looks good after fine-tune.
- well written, easy to read and follow.

**Weaknesses:**

- Statement
  - Line 27-31
    - Although the manuscript does acknowledge certain NeRF-based SLAM methods for their ability to provide dense 3D reconstruction, relevant references [1,2,3] are conspicuously absent. Additionally, some NeRF-based SLAM methods that utilize prior for pose estimation have been glossed over.
  - Line 34
    - The cited works do not adequately represent the body of literature since both methods focus exclusively on depth map estimation. This oversight should be rectified by including relevant citations [4,5].
 - Line 143
   - wrong format


- Motivation
  - The motivation for introducing single view generalizable nerf for relative pose estimation is unclear and wired.
    - Line 142
    - the definition of the x_t. In equ2, x_i is the sampled points along the ray. Does x_t is the S(r) in equ2?
    - while pixelnerf can use only a single view to render a novel view, it is much like a mono-depth estimation module. Without optical flow, an alternating solution uses iNeRF to align two single views point cloud/nerf, minimizing photometric loss. However, even most SOTA mono-depth estimation methods are insufficient to estimate accurate relative pose due to cross-view scale misalignment. Some unposed/posed Nerf only use mono-depth as a reg term. This is why the proposed method **HAS TO** introduce the out-of-shelf optical flow estimation for pose estimation. If the single view pixelnerf is removed from the proposed method, optical flow, as the correspondences, is enough to estimate the essential matrix/relative pose between two neighbor frames. Furthermore, the deep-sfm module, like banet, droidslam, deepv3d, all of these methods take multiview images to estimate both multiview depth map and the camera pose at the same time. The author should carefully address this issue during the rebuttal.
  - the motivation of flow confidence weights.
    - Line 143
    - This part could be easily solved by bi-direction consistency check with optical flow. Introducing another pretrained network for weight prediction will make the generalization issue worse.
    - Figure 9 does not show the results with dynamic objects.

  - Line 178
    - RGB VO/SLAM confronts scale ambiguity problem. The proposed method doesn't handle it at all.

  - Line 194
    - where is the definition of N, J?

  - Line 203 Equ8
  - The mono-depth is enforced by this loss term during fine-tune. As mentioned above, the optical flow for relative pose estimation is enough.
  - This loss term does not handle singular solutions.

- Experiment
  - Line 211
    - the training detail is missing.
    - the proposed method cannot run on a long video clip.
    - Most of the experiments show the object-center case. It will be great to see the results on the inside-out case either.
  - Although the authors mention that they do not claim sota accuracy of pose estimation, the metrics of the pose are still necessary to be reported on the main paper, including droid slam, droid vo, orb slam, etc.
  - Line 224
    - test-time optimization on the weight of CNN and MLP on a single video clip in a self-supervised manner is extremely hard. The problem will become worse if the precomputed optical flow is removed. The author should provide more detail about this section.

  - Line 250
    - more baseline methods are required, such as nicer-slam[2], dim-slam[3], droid-slam.
    - some unposed nerf methods with mono depth estimation module, such as nope-nerf[6]
    - and another test-time optization method, such as CVD[4], RCVD[5].

  - The pose trajectories in the figure are misaligned in both the main paper and the supplementary, which are very confusing.

- reference
  - [1] Sucar, E., Liu, S., Ortiz, J., & Davison, A. J. (2021). iMAP: Implicit mapping and positioning in real-time. In Proceedings of the IEEE/CVF International Conference on Computer Vision (pp. 6229-6238).
  - [2] Zhu, Z., Peng, S., Larsson, V., Cui, Z., Oswald, M. R., Geiger, A., & Pollefeys, M. (2023). Nicer-slam: Neural implicit scene encoding for rgb slam. arXiv preprint arXiv:2302.03594.
  - [3] Li, H., Gu, X., Yuan, W., Yang, L., Dong, Z., & Tan, P. (2023). Dense RGB SLAM with Neural Implicit Maps. The Eleventh International Conference on Learning Representations(ICLR)
  - [4] Luo, X., Huang, J. B., Szeliski, R., Matzen, K., & Kopf, J. (2020). Consistent video depth estimation. ACM Transactions on Graphics (ToG), 39(4), 71-1.
  - [5] Kopf, J., Rong, X., & Huang, J. B. (2021). Robust consistent video depth estimation. In Proceedings of the IEEE/CVF Conference on Computer Vision and Pattern Recognition (pp. 1611-1621).
  - [6] Bian, W., Wang, Z., Li, K., Bian, J. W., & Prisacariu, V. A. (2023). Nope-nerf: Optimising neural radiance field with no pose prior. In Proceedings of the IEEE/CVF Conference on Computer Vision and Pattern Recognition (pp. 4160-4169).

**Questions:**

I am happy to adjust the rating if the author properly addresses the concerns mentioned during the rebuttal.

----

after rebuttal:

I adjust the rate to the positive side.

**Limitations:**

Yes.

---

> ### Author Rebuttal · Authors · 2023-08-09
>
> ### iMAP and robust unposed NeRF
> We added iMAP methods to the related works and NeRF methods. Note that iMAP addresses a separate problem: our contribution enables end-to-end training of pixelNeRF without precomputed poses, whereas iMAP performs offline, gradient-descent based pose and NeRF optimization.
> ### Missing references to Consistent Video Depth Estimation and Robust Consistent Video Depth Estimation
> We will add CVD and RCVD to our related work. CVD/RCVD fine-tune a pre-trained depth network for temporal consistency, whereas we train a generalizable 3D scene representation; they are not significantly related.
>
> ### Typo on line 143
> Thanks - fixed!
>
> ### pixelNeRF for pose estimation is weird
> We use the same network for pose estimation and rendering since a) it forces our pixelNeRF to learn good geometry in order to predict correct poses b) it avoids scale inconsistency between poses and rendering. As for why we involve geometry in pose estimation, instead of a 2D solver, see our response below on that point. We can also use a separate depth network instead of re-using pixelNeRF; see Tab. 2, “Depth Regression” of the response PDF to see that it works almost as well.
>
> ### 142 ambiguity
> x_t and x_i both refer to 3D points, where x_i in eq2 refers to the 3D point at the i’th sample along the ray, and x_t refers to the 3D point observed at pixel p_t. We will amend the text accordingly to make this notation more explicit.
>
> ### iNeRF formulation
> iNeRF requires a pre-trained NeRF for pose inversion, which we do not have, and requires many gradient descent steps to align the model for accurate poses, which we cannot afford to embed in the training loop.
>
> ### 2D correspondences sufficient for pose estimation
> Solving for the essential matrix from 2D correspondences introduces scale ambiguity between the 3D scene representation and estimated poses: the pose from the 2D correspondences is extremely unlikely to be the scale appropriate for the pixelNeRF’s constant near and far plane. See Tab. 2, column “2D-Only Pose Solver” of the response PDF; renderings catastrophically fail with this approach.
>
> ### DROID-SLAM uses just 2D correspondences
> Using any of these methods incurs the same scale ambiguity between generalizable neural scene representation and pose estimation described above and in the overview text. Also note that none of these methods solve for a 3D scene representation, but rather solve for depth maps and poses.
>
> ### Flow confidence weights; bidirectional flow check simpler; generalization is now worse
> See our overview discussion on flow confidence weights and dynamic object masking. We ablate this bidirectional flow check in Tab. 2 (“Bidirectional Consistency Flow Weights”) of the response PDF and report a significant decrease in rendering quality. Also consider this estimation asks how similar two features are and tends to generalize well.
>
> ### Proposed method doesn't resolve scale ambiguity
> Please see the overview text for expanded discussion on the scale ambiguity we address, which is between the poses and 3D scene representation, and how our method resolves it by using the same geometry estimation for pose estimation and rendering.
>
> ### Missing definitions (194,203)
> Thanks! We’ll amend the text to be more explicit. N refers to the number of timesteps or frames and J refers to the subset of context images used for rendering.
>
> ### Scale resolved with fine-tuning
> The fine-tuning mechanism is not involved during training of the scene representation; see the overview discussion on fine-tuning and scale ambiguity. We cannot afford to perform fine-tuning during the forward pass.
>
> ### Our loss term does not handle singular solutions
> If there is a singular solution due to textureless images, it should not affect the rendering or 3D scene representation, and is not a priority since our interest is the 3D scene representation. We will add this discussion to our limitation section.
>
> ### Sliding window experiments details; Cannot run on long sequences
> The sliding window method is an offline extension for explicitly chaining predicted camera poses on subsequent video subsequences to accommodate longer sequences. There is no optimization when using the sliding window approach; we simply query our trained model (such as on CO3D).
> While our method does not accommodate long sequences, the ~30 frames with considerable frameskip we use is within the pose distribution typically used for training such 3D scene representations.
>
> ### Inside-out scenes
> Only CO3D results are outside-in; please see are results on RealEstate10K and KITTI which are inside-out.
>
> ### Comparison with ORB-SLAM and DROID-SLAM
> We compare with ORB-SLAM and DROID-SLAM in Tab. 1a of the author response PDF and Fig. 6 of the supplemental PDF. We outperform both ORB-SLAM and DROID-SLAM in this setting.
>
> ### Test time optimization of CNN and MLP difficult and expensive
> Please see the overview discussion on the application of fine-tuning, which is not core to our method pipeline. Note that a) the fine-tuning results we demonstrate are on our pretrained model, not from scratch, b) the optimization is empirically robust, and c) it is not considerably more expensive to fine-tune the image features as well.
>
> ### DROID-SLAM/Dim-SLAM/NoPe-NeRF/RCVD baselines
> See Tab. 1 and Fig. 1b for comparison to NoPe-NeRF and Tab. 1a of the author response and Fig. 6 of the supplement for comparisons to DROID-SLAM. Since CVD aims to fine-tune a monocular depth estimator to be temporally consistent for a single video, instead of training a 3D scene representation, we respectfully disagree with CVD or RCVD being apt comparisons, but if you feel otherwise we can discuss it and evaluate the comparison.
>
> ### Misaligned pose plots
> We assume you refer to the reversed captions of figure 4, which we will amend in the paper revision, but otherwise if you can be more specific about which poses are misaligned we will be happy to correct it.

---

> > ### Comment · Reviewer_52at · 2023-08-19
> >
> > Thanks for the rebuttal. Although I still have concerns about the `scale drifting`, which is not the `scale ambiguity issue responded to by the author, most concerns are properly addressed during the rebuttal.
> >
> > Some missing Nerf-slam-based methods still need a proper reference.

---

> > > ### Author Response · Authors · 2023-08-21
> > > **Follow-up response on scale drift**
> > >
> > > Thanks for responding and again for the detailed review, as well as for the the clarification on the type of scale discrepancy referenced. Since our method makes frame-to-frame estimates, as opposed to performing multi-frame optimization, indeed scale drift error could potentially accumulate in the feedforward setting. Adding a scale parameter to the Procrustes could address this, but we found that forcing the model to use a single scale yielded more stable training (Tab. 2 of author response PDF). Enforcing a single scale also has the benefit of  more temporally consistent depth estimates, as opposed to the scale-invariant depth estimated commonly by monocular depth networks which are notoriously difficult to normalize correctly. Multi-frame feedforward optimization is exciting future work.
> > >
> > > We will make sure to add the NeRF-SLAM methods mentioned.

---

### Official Review · Reviewer_kV1r · 2023-06-30

**Soundness:** 3 good
**Presentation:** 3 good
**Contribution:** 4 excellent
**Rating:** 7
**Confidence:** 4

**Summary:**

This paper proposes a method to address the challenge of reconstructing 3D neural fields from images and learns them in a self-supervised manner. The main contribution of this method is the joint reconstruction of camera poses and 3D neural scene representations within a single forward pass.

**Strengths:**

1. This paper achieves state-of-the-art performance in its specific setting.
2. The writing in this paper is clear.
3. This paper addresses an important problem. It overcomes the previous limitation in training generalized Neural Radiance Fields that require posed videos. It enables direct training on unposed video data.
4. This paper is innovative in its methodology and no one has utilized similar approaches to address this problem before.

**Weaknesses:**

1. The paper lacks clarity in explaining the ablation study, such as the incomplete caption for Table 3. Moreover, the abbreviations used in the table are not defined, making it unclear what they refer to.
2. The paper lacks ablation experiments on not learning intrinsic camera parameters.

**Questions:**

1. I cannot understand how this paper addresses the generalization issue. Why does PixelNeRF have good generalization capabilities in monocular depth prediction?
2. How does this paper address the problem of scale ambiguity in monocular depth estimation?

**Limitations:**

This paper discusses its limitations. However, I would have expected to see more failure cases from internet data.

---

> ### Author Rebuttal · Authors · 2023-08-09
>
> ### The ablation study is unclear in its abbreviations and references
> We agree and have updated the main paper’s ablation table with expanded ablation references as well as its corresponding caption and experiment text. Please see author response PDF Tab. 2 for expanded reference names and caption. In case it is useful for correspondence, note that in the submission’s ablation table “MLP-Pose” refers to pose regressed by an MLP (as opposed to using a solver); “No Flow Weights” refers to using the pose solver but not using the confidence weights to guide the optimization; “Full” refers to the full model of using the solver with the confidence weights.
> ### This paper lacks ablations on not learning intrinsics parameters
> In this paper we use intrinsics parameters when known and only predict them otherwise, such as on the YouTube videos where no such information is available. We noted this on line 93 but we will make this more explicit in the experiments section as well. We also ran an additional ablation experiment on using constant and predicted intrinsics, reported in Tab. 3 of the author response PDF. Surprisingly, using constant intrinsics performs similarly to using SfM-estimated intrinsics, and predicting them actually outperforms using the SfM-estimated intrinsics. While this is surprising, one reason could be that the dataset intrinsics are not perfect but estimated by SfM or that our model may be predicting intrinsics which better correspond to its geometry estimate.
> ### Not sure how this method addresses the generalization issue of pixelNeRF
> PixelNeRF has been shown in several instances to be a robust geometry estimator, used for depth estimation (see “Behind the Scenes: Density Fields for Single View Reconstruction”, e.g.). It has been demonstrated that pixelNeRF generality scales favorably when increasing dataset size  (see “Objaverse-XL: A Universe of 10M+ 3D Objects”). In general, depth estimation is a subset of the 3D geometry that pixelNeRF estimates. Our key contribution is a step towards training of pixelNeRF on internet-scale, uncurated video datasets. This would dramatically improve the generality of pixelNeRF, as the training distribution could now essentially be all natural videos on the internet. While some steps towards that goal are outstanding (such as addressing dynamic 3D reconstruction), we believe that our paper takes a significant step towards that goal.
> ### How does this paper address the scale ambiguity in monocular depth estimation?
> Most monocular depth estimators output depth in an arbitrary scale, and normalizing their outputs is notoriously difficult, even for data from the same domain (such as driving scenes). The reason for this is related to the scale ambiguity we describe in the overview discussion, which boils down to inconsistent scale of camera poses from different videos. More specifically, the depth targets used to train those depth estimators are often only known up to scale, which makes training accordingly challenging when treating those as metric targets. Our model naturally resolves this scale ambiguity by predicting camera poses using our model’s rendering geometry. That is, since our camera poses are a function of our model’s predicted geometry, they are inherently of the same scale, removing any ambiguity introduced by depending on SfM per-scene optimized poses.
> We assume that is the ambiguity you are referring to, but another ambiguity you might be referencing is the inherent scale ambiguity between our scene representation and the metric scale of the real world, which is impossible without a metric reference (such as RGBD). And lastly note that the scale ambiguity we refer to addressing in the text and the overview discussion is rather the difference in scale between the estimated poses (traditionally estimated via COLMAP) and the generalizable scene representation (which is regressed during training). See the overview discussion for an elaboration on this ambiguity and how it otherwise poses a significant challenge for training 3D scene representations such as pixelNeRF, which we overcome.
> ### Would expect to see more failure cases on internet data
> Indeed, since we have not trained our model on a large-scale internet dataset, our model does not generalize perfectly yet to random internet videos, for reasons which include non-robust intrinsics prediction and out-of-distribution geometry estimation. Note that we train our pixelNeRF on the same scale of datasets typically used to train pixelNeRF (CO3D, e.g.), and just aim in these experiments to reproduce the same level pixelNeRF, without depending on precomputed poses, rather than increase generalization. The point made in our presentation of fine-tuned results for a few internet videos was to simply show that we can optimize on in-the-wild videos, and that although we do not yet scale up training to internet-scale data (see outstanding bottlenecks mentioned above), our model takes a large step towards enabling such internet-scale training. We will further include an additional set of results in the supplemental material of our model on random internet videos.

---

> > ### Comment · Reviewer_kV1r · 2023-08-18
> >
> > Thanks for the rebuttal, which has well addressed my concerns. So I keep the positive side.

---

### Official Review · Reviewer_3ZaF · 2023-07-06

**Soundness:** 3 good
**Presentation:** 3 good
**Contribution:** 3 good
**Rating:** 5
**Confidence:** 5

**Summary:**

This paper proposes a general method for 3D neural scene reconstruction and camera pose estimation from a video sequence. The method takes a set of video frames as input and outputs the re-rendered video frames and estimated camera poses. The method is based on PixelNeRF, which is a general NeRF method that takes extracted image features as input.

The method first uses single-view PixelNeRF to compute per-frame point clouds. Then, the frame-to-frame 3D scene flow is estimated by leveraging the estimated 2D optical flow. The 3D scene flow is then used to constrain the frame-to-frame camera poses, which are solved by minimizing a weighted least-squares problem. Two losses are used to supervise the training: the RGB loss, which forces the PixelNeRF-based rendered images to be close to the input images; and the flow loss, which makes the projection offset of neighbor frames' point clouds close to the 2D optical flow.

**Strengths:**

1. A nice solution to learn a general NeRF representation for multiview images without camer pose information.
2. The quality of the rendering results for testing multi-view images are impressive.

**Weaknesses:**

The experiments can be improved to further verified the advantage of the proposed method:

1. Only one quantitative pose estimation comparison is shown, there is no quantitative comparisons with BARF. Besides, Nope-NeRF is a similar method that takes sequent frames as input for pose estimation, it better have one more comparison with Nope-NeRF.

2. The quantitative results in Tab 1 are fine-tuned or not? It should be better to show both results.

**Questions:**

Overall,  the proposed method is reasonable, but I do have questions:

1. There should be a scale problem when estimating the per-view point cloud by single-view pixelNeRF. It seems that the Eq (4) didn't consider it, why?

2. The confidence weight is output by a network. Therefore, in my understanding, this network can implicitly handle the occlusion and specular by downweighting corresponding flow. But it seems that the Masked RGB images shown in Fig 9 mask out too much content (even for textured diffuse regions). Is this because of the inaccurate 2D optical flow estimation?

**Limitations:**

The proposed method learns a conditioned generaI NeRF network for multiview images. It is still limited since the images should be kept to accompany the learned model.

---

> ### Author Rebuttal · Authors · 2023-08-09
>
> ### Only one quantitative pose estimation is reported, and we should compare with a more recent unposed NeRF method (such as NoPe-NeRF)
> Please note that we extensively benchmark with the appropriate baselines of RUST and VideoAutoencoder, including quantitative evaluations of pose estimation on each dataset in Tab. 2 of the main paper. We also include comparisons to ORB-SLAM in Tab. 2 of the main paper and add DROID-SLAM quantitative comparisons in Tab. 1a of the author response, as well as qualitative pose estimation results to both methods in Fig. 6 of the supplemental PDF. We outperform all baselines considered for pose estimation on these difficult sequences.
> As for the unposed NeRF family of methods as baselines, note that their goal is orthogonal to ours, in that they aim to optimize a NeRF for a single scene without poses, whereas we are interested in training a generalizable NeRF representation without poses. See the general overview response for further discussion on this point. With that distinction made, we agree that our experiment would be stronger with a more recent and robust unposed NeRF method. See the author response PDF, Fig. 1b and Tab. 1b for quantitative and qualitative evidence that we outperform both BARF and NoPe-NeRF for pose estimation. We will include these results in the experiments section of the main paper.
> ### Fine-tuned results ambiguity
> In general, all results in the paper are zero-shot (not fine-tuned), unless explicitly marked as fine-tuned. We will clarify this distinction in the experiments section of the main paper. Also note that the fine-tuning mechanism is not part of our model pipeline, which aims to remove precomputed poses from the training process of generalizable 3D representations, but rather a demonstration that our self-supervised formulation allows us to quickly optimize on any scene for higher-quality novel view synthesis. See the overview discussion for further elaboration on this point.
> ### Scale ambiguity in single-view depth estimation that could have been accounted for in Procrustes estimation
> This is an insightful point and while your analysis is correct, in practice we omit it to encourage the model to learn a consistent scale, and observe that the model’s depth estimates are generally of the same scale. Surprisingly, we found that constraining the scale to be the same across the video is critical and the model performance suffers with the scale degree of freedom. See Tab. 2 (column “Scale Adjusting Procrustes”) of the author response PDF for an explicit evaluation demonstrating this.
> ### Confidence maps are concerningly sparse, even discounting textured diffuse regions
> This is an acute observation and the weights are indeed relatively sparse, but note that our model theoretically only needs four points to solve for correct pose, and our model can therefore afford to choose fewer but more accurate correspondences. Also note that our model can also mask out parts of the scene where it is not confident about the geometry, such as sky or distant regions, not just bad correspondences. Please see Fig. 1a of the author response PDF for an illustration of our model’s flow weight mask including dynamic objects, as well as the overview discussion for further elaboration on the flow weights.
> ### The method is limited since the images should be kept to accompany the learned model
> While image-conditioned scene representations, such as pixelNeRF, keep the images during rendering, note that all NeRF-based representations keep around some representation, whether it be the weights of a large global network or the densities and radiance of a voxel grid. Also note that one could potentially exchange an image-conditioned scene representation to a voxel-grid based one via sampling the 3D radiance field at the voxel locations, or alternatively distill the representation into a global network. We agree nonetheless that optimizing a global NeRF, such as BARF, and not keeping keyframes is an exciting and self-contained direction for future work.

---

> > ### Comment · Reviewer_3ZaF · 2023-08-21
> >
> > Thanks for clarifying my questions. Please incorporate the materials in the rebuttal into the revision

---

### Official Review · Reviewer_oZ25 · 2023-07-06

**Soundness:** 3 good
**Presentation:** 3 good
**Contribution:** 3 good
**Rating:** 6
**Confidence:** 4

**Summary:**

The paper proposes using scene flow to optimize for camera poses to produce generalizable 3D radiance field. The key contribution is the joint optimization of the camera poses and 3D neural scene representations in an single forward pass. The method has been evaluated on multiple datasets and performs well on datasets where traditional pose estimation fails.

**Strengths:**

The motivation for using an joint optimization of camera poses and reconstruction has wide applications and easy to scale.

As proposed by the current method multiple views of a scene inherently produce good correspondences, which leads to reasonable optical flow estimates. Using and incorporating optical flow into the 3D pose estimation and reconstruction helps overall reconstruction pipeline.

Using optical flow for reconstruction and Neural radiance fields without cameras poses have been well independently. Combining them into a single formulation is still less explored. the current method lifts the optical flow to scene flow using neural scene representations and uses them to optimize. This is interesting direction to explore.

The results shows that the method is able to perform better than traditional pose estimation methods like ORB-SLAM in some of their failure cases.

**Weaknesses:**

The current method proposes a tradeoff by stating instead of using camera poses first and neural radiance field later as a two stage process, joint optimization is a more promising direction. but using joint optimization adds additional optimization constraints which have not been extensively discussed. It would be nice to see what are the failure cases of using the joint optimization compared to the two-stage optimization process.

The camera pose estimation has only been evaluated with ORB-SLAM3 or VidAE both of which do not incorporate neural radiance fields for 3D pose estimation. Recently there have been better methods which optimize for both radiance fields and pose simultaneously. having such baseline will give a better understanding of the accuracy of the algorithm.

The number of frames used for error computation is very small. i.e. 20 frames and 200 frames. The advantage of the method states that the joint optimization is beneficial to scale easily. Showing the method accuracy compared to orb on larger datasets or sequences like

**Questions:**

in Fig3 results were shown using BARF for pose estimation, why are there no quantitative results using the same?

**Limitations:**

Limitations have been well discussed.

---

> ### Author Rebuttal · Authors · 2023-08-09
>
> ### Pose estimation is only compared with non-NeRF based methods, and unposed NeRF methods have been proposed which yield better results with a more accurate comparison
> We add a NoPe-NeRF comparison (see author response PDF, Fig. 1b and Tab. 1b); we succeed where BARF and NoPe-NeRF fail. Also recall that the unposed NeRF family of methods has an orthogonal goal to ours, in that those methods jointly optimize poses and NeRF for a single scene, whereas our aim is to train a generalizable scene representation.
>
> ### Pros and Cons compared to two-stage approach, failure cases of proposed joint optimization of poses and NeRF
> Using a two stage approach usually involves an non-differentiable pose estimator, either running in near real-time or offline. For the offline two-stage approach, such as using Colmap, the benefits include that a) it’s less expensive, b) there exists a more developed ecosystem of optimization tools for it, such as intrinsics solvers and loop closure mechanisms, and c) it can optimize on longer sequences with greater accuracy. The drawbacks are that it is prohibitively slow and can not be embedded in a real-time training loop and is known to still regularly fail, particularly on rotation-dominant sequences. Online non-differentiable systems, such as ORB-SLAM, have the benefit of being fast enough to run alongside the training of a downstream task, with the main drawback being that they often fail due to their lack of a scene prior. See Tab. 1a of the author response PDF where we show that ORB-SLAM loses tracking on over half of the CO3D sequences.
> Joint optimization methods have the benefit of being differentiable, meaning we can resolve the scale ambiguity between poses and renderer (discussed extensively in overview discussion), are real-time, and improve with the rendering loss as well. The main failure mode is that the poses are less accurate than an offline-optimization method, such as Colmap, and the renderings are proportionately blurry to the pose inaccuracy. Please see the supplemental document (Figures. 2 and 6) for blurred renderings and non-perfect pose estimations. Joint optimization further also incurs additional cost to a two-stage approach, as we have to additionally estimate geometry and poses for each timestep in each forward pass.
>
> ### Number of frames is very small, which contradicts the claim that our contribution is to scale up training
> Note we are referring to the goal of scaling up the size of the datasets, e.g. moving from Co3D or ShapeNet  to all of YouTube, rather than scaling up the length of the videos. Though our experiments indeed usually span 20-30 frames instead of the 1000 frames used for odometry evaluations, note that we often use a considerable frameskip between images, and more sophisticated pose estimation algorithms in future work will likely be able to accommodate even larger frame skips. For instance, the 200 frames cited for the Tanks and Temples are downsampled from the original video to just 1 FPS, spanning a full circle around the Excavator, and that CO3D long-trajectory evaluations also use a significantly low framerate, such that the 200 frames typically similarly circles the full object. The resulting camera baseline from our framerate downsampling is appropriate for our goal of training a generalizable pixelNeRF and within the pose distribution typically used to train these methods.
>
> ### BARF comparisons should have quantitative pose evaluations
> We have added quantitative comparison to BARF and NoPe-NeRF (Tab. 1b of author response PDF).

---

### Author Rebuttal · Authors · 2023-08-09

We appreciate the time and energy the reviewers have invested in reviewing our paper and for offering insightful and constructive feedback, which will make our paper clearer and stronger. We are glad that reviewers recognized our paper’s contribution in addressing an “important problem” (kV1r) in training a “general NeRF representation without camera pose information” (3ZaF) which has “wide applications and is easy to scale” (oZ25) and “achieves state-of-the-art performance in its specific setting” (kV1r).

## General Clarifications
### Paper goals and their difference to conventional Odometry & SLAM methods.


Our goal is to build a neural network that can jointly predict a 3D radiance field as well as camera poses in a single forward pass, without the need of off-line pre-processing training videos with Structure-from-Motion. We succeed at this goal and train pixelNeRF end-to-end on challenging real-world video with significant camera rotation.

On the challenging CO3D 10-Category video dataset, our method obtains more accurate poses than ORB-SLAMv3 and DroidSLAM (see author reply, Table 2a). Thus, we do claim - and demonstrate - that training pixelNeRF with poses obtained from either DroidSLAM or ORB-SLAMv3 would perform worse than our method. However, we explicitly (Lines 78, 219) do not claim that our method is competitive with offline Structure-from-Motion, such as Colmap, especially on long-horizon pose estimation, where loop closures and global pose graph optimization are critical. However, this is irrelevant to our goal of removing the SfM pose bottleneck from training these methods. To illustrate this point of the computational infeasibility of scaling up SfM methods, consider that the CO3D dataset totaled over 5 GPU years to estimate poses for, and CO3D is an insignificant fraction of the size of internet-scale datasets.

### Fine-tuning mechanism and comparison with single-scene unposed NeRF methods, such as BARF and NoPe-NeRF.
While our key contribution is training of pixelNeRF with a feedforward pose estimate, the fine-tuning serves to demonstrate that for longer sequences, our self-supervised loss yields gradients that lead to high-quality novel view synthesis within minutes. We have added comparisons to both BARF and NoPe-NeRF to Section 4; please see the author response PDF, Fig. 1b and Tab. 1b. BARF and NoPE-NeRF both fail to capture the rotation-dominant pose distribution; our method outperforms both of them.

### Scale ambiguity
The scale ambiguity we refer to resolving is the discrepancy between the world scale estimated by the offline pose-estimation method (usually COLMAP), and the world scale estimated by the generalizable 3D scene representation. To train these 3D scene representations, it is assumed that there is a consistent scale of the poses which the scene representation can learn to predict geometry in. Several view synthesis methods (GenVS, GPNR, Diffusion with Forward Models) introduce their own nontrivial scale-normalization steps and note that they are critical for stable training.

## Additional Experiments
In addition to reviewer-specific experiments, we highlight a few experiments common to all reviewer requests below.
### More robust and recent unposed NeRF methods (NoPe-NeRF)
We added a NoPe-NeRF comparison in addition to BARF; see Fig. 1b and Tab. 1b of the author response PDF. Despite NoPe-NeRF’s monocular depth prior addition to BARF, both do not capture the correct pose distribution.
### Scale ambiguity demonstration
To demonstrate the difficulties involved with the mentioned scale ambiguity, we randomly scale the poses estimated by our model before rendering and report the corresponding decrease in rendering quality in Tab. 4 of the author response PDF.
### Confidence weights clarification and additional demonstration
As reviewer 3ZaF mentions, the flow weights that our model estimates are quite sparse. Since our pose estimation theoretically only requires 4 3D correspondences, our model can afford to be selective for good correspondences. Also note that our model not only masks out bad correspondences and nonrigid scene elements, but also areas where it has geometry uncertainty, such as in the sky region. Note that this flow weighting is critical to robust pose estimation, which we ablate in Tab. 2 of the author response PDF (“No Flow Weights” column). We include an instance in Fig. 1a of the author response PDF where our model indeed masks out a dynamic object (a bicyclist), but note that because the flow weights are relatively sparse, we are not suggesting that our model produces a tight “dynamic object mask”, but rather mention dynamic objects being masked to provide intuition about why the flow weights are critical.

---

### Decision · Program_Chairs · 2023-09-21

**Decision:**

Accept (poster)

**Comment:**

The paper proposes using scene flow to optimize for camera poses to produce a generalizable 3D radiance field. The key contribution is the joint optimization of the camera poses and 3D neural scene representations in a single forward pass. After rebuttal, all reviewers recommended acceptance. The key strengths are good paper writing, a significant novelty in combining optical flow with NeRF, and impressive results. AC agrees with the reviewers and recommends acceptance.